Microbial communities respond to experimental warming, but site matters

Cregger Melissa A. 1 mcregger@utk.edu
Sanders Nathan J. 1 2
Dunn Robert R. 3
Classen Aimée T. 1 2
1 Department of Ecology and Evolutionary Biology, University of Tennessee , Knoxville, TN , USA
2 The Natural History Museum of Denmark, University of Copenhagen , Copenhagen , Denmark
3 Department of Biological Sciences, North Carolina State University , Raleigh, NC , USA
Stanford Jack
Electronic publication date: 2014 Apr 24
Publication date: 2014
Volume: 2
Electronic Location ID: e358
Received 2013 Dec 4; Accepted 2014 Apr 2
Copyright: © 2014 Cregger et al.
Copyright year: 2014
Copyright holder: Cregger et al.
License: This is an open access article distributed under the terms of the Creative Commons Attribution License, which permits unrestricted use, distribution, and reproduction in any medium, provided the original author and source are credited.
License URL: https://creativecommons.org/licenses/by/3.0/

Keywords: Decomposition, Microbial communities, Global warming, Soil enzyme activity, Eastern deciduous forests, Ecosystem function

Funding: U.S. DOE PER DEFG02-08ER64510 National Science Foundation Dimensions of Biodiversity NSF-1136703 DOE GREF Fellowship Department of Ecology and Evolutionary Biology at the University of Tennessee Funding was provided by U.S. DOE PER award (DEFG02-08ER64510) and a National Science Foundation Dimensions of Biodiversity grant (NSF-1136703) to NJ Sanders and RR Dunn. M Cregger was supported by a DOE GREF Fellowship and the Department of Ecology and Evolutionary Biology at the University of Tennessee. The funders had no role in study design, data collection and analysis, decision to publish, or preparation of the manuscript.

==============================
Because microorganisms are sensitive to temperature, ongoing global warming is predicted to influence microbial community structure and function. We used large-scale warming experiments established at two sites near the northern and southern boundaries of US eastern deciduous forests to explore how microbial communities and their function respond to warming at sites with differing climatic regimes. Soil microbial community structure and function responded to warming at the southern but not the northern site. However, changes in microbial community structure and function at the southern site did not result in changes in cellulose decomposition rates. While most global change models rest on the assumption that taxa will respond similarly to warming across sites and their ranges, these results suggest that the responses of microorganisms to warming may be mediated by differences across the geographic boundaries of ecosystems.

Introduction

Soil microbial community structure and function is directly regulated by temperature and indirectly by temperature effects on the aboveground plant community, thus global warming may rapidly and dramatically alter the structure and function of soil communities. Indeed, experiments demonstrate that warming alters microbial communities (Frey et al., 2008; Schindlbacher et al., 2011; Zogg et al., 1997). Microbial responses vary among experiments in ways that can seem idiosyncratic, which may result from the extent of warming, historical background climate where the experiments are conducted, or due to differences in the biotic aboveground community. Most warming studies to date have employed straightforward experimental designs with two levels of experimental warming—warmed vs. ambient—and implemented those treatments at a single site (Lu et al., 2013; Rustad et al., 2001; Wolkovich et al., 2012). Such designs, though informative, can limit the ability to predict responses to warming under different temperature regimes or geographic variation in community responses.

Here, we examine the extent to which microbial communities and their function respond to warming and if these responses differ between sites with different climatic regimes. We warmed soil communities at the northern and southern extremes of the range of Eastern deciduous forests in the United States. Both sites have an array of open-top chambers (OTCs) that are warmed in a regression design (Cottingham, Lennon & Brown, 2005) boosting temperatures from 1.5 to 5.5 °C above ambient in ∼0.5° steps. This design allows us to assess the functional responses of microbial communities across a variety of temperatures at two locations. Work at these sites and elsewhere indicates that the effect of temperature on animal communities varies geographically (Pietikainen, Pettersson & Baath, 2005; Tewksbury, Huey & Deutsch, 2008). Across taxonomic groups, the abundance of species operating near their critical thermal maxima at low latitudes tends to be regulated more by warming than that of species operating farther from their critical thermal maxima. Thus, warming may have larger effects on species, and their functions, at warmer low-latitude sites than at cooler high-latitude sites. Previous work suggests that while bacterial and fungal diversity and abundance might increase in response to warming at low background temperatures, at high temperatures bacteria may respond positively, but fungi either fail to respond or respond negatively to warming (Pietikainen, Pettersson & Baath, 2005). This differential response may affect the rate at which carbon and other nutrients are cycled in ecosystems. Bacteria tend to degrade more simple carbon substrates (de Boer et al., 2005) thus initially there may be an increase in CO2 efflux into the atmosphere followed by a leveling off as more recalcitrant carbon is left in the soil.

Given the different climates, particularly with regard to temperatures at the two experimental sites, and that soil community structure and function are temperature sensitive, we predicted that microbial communities would respond to warming, but these effects would differ between the two sites and depend upon the amount of warming.

Materials and Methods

This experiment is described in Pelini et al. (2011). Briefly, 12 octagonal (5 m diameter) OTCs were established in 2009 and activated in January 2010 at a southern site (Duke Forest, 35°52′0″N, 79°59′45″W) and a northern site (Harvard Forest, 42°31′48″N, 72°11′24″W). Each chamber has a ±20 cm oak tree in the center of the chamber to serve as a thermal storage mass in order to avoid a cold core in the middle of the chamber. Three chambers serve as unheated controls, and the remaining nine chambers manipulate air temperature in 0.5 °C increments using hydronic heating and forced air from 1.5 °C to 5.5 °C above ambient. Target and observed temperatures are strongly correlated (r2 = 0.99). Mean annual air temperature was 15.5 °C and mean annual precipitation was 1140 mm at the southern site and 7.1 °C and 1066 mm, respectively at the northern site (Pelini et al., 2011). The southern site was established in a mixed deciduous, 80 year-old oak-hickory (Quercus alba-Carya sp.) forest, with an understory that was dominated by oak (Quercus alba), red maple (Acer rubrum), and hickory (Carya sp.). The soils are mainly Ultic Alfisols with a soil pH, as measured in calcium chloride (Carter & Gregorich, 1993), of 3.5 ± 0.03. The northern site was established in a mixed deciduous, 70 year-old oak-maple (Quercus rubra-Acer rubrum) forest with an understory that was dominated by blueberry (Vaccinium sp.), pine (Pinus strobus), and maple (Acer pensylvanicum). The soils are mainly of the Canton series (coarse-loamy over sandy or sandy skeletal, mixed, semi-active, mesic Typic Dystrudepts) (Melillo et al., 2011) with a soil pH, as measured in calcium chloride (Carter & Gregorich, 1993), of 3.6 ± 0.08. Previous research on ants at these sites showed that ant forager abundance and richness correlated to experimental temperature increases at the southern site, but not the northern site. Further, individual ant species responded differently to temperature increases at the southern site (Stuble et al., 2013). Another approach to examining whether the responses of microbial communities differ among regions would consist of installations of this warming experiment at multiple sites from Duke Forest to Harvard Forest rather than at only two sites. However, such a design is currently cost prohibitive (both in terms of setting up the experiment and in processing samples). But, importantly, our design allows us to explore the dynamics of communities at range boundaries, where it is predicted that the strongest responses to ongoing warming will be Parmesan & Yohe (2003) and Walther, Berger & Sykes (2005).

We monitored air temperature as well as organic and mineral soil temperature continuously in each chamber with Apogee data loggers (model SQ110; Apogee Instruments Inc., Logan, UT, USA). Relative humidity (HS-2000V capacitive polymer sensors; Precon, Memphis, TN, USA) and soil moisture (Model CS616 TDR probes; Campbell Scientific Inc.) were also continuously monitored in each chamber at both sites. Monitored air temperatures within the chambers matched the target temperatures (Burt et al., in press). Soil temperature in the organic and inorganic layers was positively correlated with air temperature whereas soil moisture was never correlated with air temperature (Burt et al., in press).

Five soil cores (2-cm diameter, 5-cm depth) were collected from within each of the 12 warming chambers on April 23rd, 2011 at Duke Forest and on May 17th, 2011 at Harvard Forest (5 cores/chamber × 12 chambers × 2 sites = 120 soil cores). We were unable to sample multiple times across the year because we needed to limit disturbance to the plots, thus we selected a time when we knew the microbial community would be actively degrading soil carbon. Soil from each chamber was homogenized (24 total samples); 15 g of soil were immediately removed from the homogenized sample, stored on dry ice in the field, and kept frozen at −80  °C until analyzed. The remaining soil was sieved (2 mm) and assayed for potential extracellular enzymatic activity and soil gravimetric water content within 48 h of collection.

To explore how warming altered microbial community structure and function we assessed microbial abundance using quantitative PCR (Castro et al., 2010), microbial community composition using terminal restriction fragment length polymorphism (TRFLP) (Cregger et al., 2012; Singh et al., 2006), the potential activity of nine extracellular enzymes, and a microbially mediated ecosystem function—cellulose decomposition.

To assess bacterial and fungal gene copy number, a commonly used proxy for abundance (Allison & Treseder, 2008), we ran quantitative polymerase chain reaction (qPCR) on each individual sample in conjunction with primers Eub 338 and Eub 518 for 16S ribosomal DNA and nuSSU1196F and nuSSU1536R for 18S ribosomal DNA (Castro et al., 2010). PCR mixtures for both 16S rRNA and 18S rRNA gene amplification contained 15 µl of SYBR green master mix (Invitrogen, Life Technologies, Grand Island, NY), 5 µmol of each primer (Eurofins MWG Operon, Huntsville, AL), and 1 µl of sample DNA diluted 1:10 in sterile water. Reactions were brought up to 30 µl with sterile water. Amplification protocol for the 16S rRNA gene consisted of an initial denaturing cycle of 95 °C for three minutes. This cycle was followed by 39 cycles of 95 °C for 15 s, 53 °C for 15 s, and 72 °C for 1 min. Amplification of the 18S rRNA gene consisted of an initial denaturing cycle of 95 °C for three minutes. This cycle was followed by 39 cycles of 95 °C for 15 s, 53 °C for 15 s, and 70 °C for 30 s. Abundance was quantified by comparing unknown samples to serial dilutions of 16S and 18S rDNA from Escherichia coli and Saccharomyces cerevisiae, respectively in each PCR run. After completion, for both ribosomal genes, a melting curve analysis was conducted to ensure purity of the amplification product. PCR amplification was performed on a 96-well Chromo4 thermocycler (Bio-Rad Laboratories, Hercules, CA).

We assessed microbial community composition using terminal-restriction fragment length polymorphism (TRFLP), which provided fingerprints of the bacterial and fungal communities (Singh et al., 2006). Due to decreases in fluorescence when samples were multiplexed, we performed bacterial and fungal TRFLPs in separate reactions. PCR was performed to amplify the 16S rRNA gene from bacteria using primers 63f (Marchesi et al., 1998) and 1087r (Hauben et al., 1997) and the fungal ITS region using primers ITS1f (Gardes & Bruns, 1993) and ITS4r (Singh et al., 2006). PCR mixtures contained 5 µl 10× KCL reaction buffer, 2 µl 50 mM MgCl2, 5 µl 10 mM dNTPs (Bioline, Tauton, MA), 1 µl 20 mg/ml BSA (Roche, location), 0.5 µl (2.5 Units) Taq DNA polymerase (Bioline, Tauton, MA), either 1 µl of each bacterial primer or 2 µl of each fungal primer (Labeled primers—Invitrogen, Life Technologies, Grand Island, NY; unlabeled primers—Integrated DNA Technologies, Coralville, IA), and 1 µl sample DNA diluted 1:10 in sterile water. All PCR reactions were performed using a 96-well Tgradient thermocycler (Biometra, Germany). DNA was amplified with an initial step of 95 °C for 5 min, followed by 30 cycles at 95 °C for 30 s, 55 °C for 30 s, and 72 °C for 1 min. This was followed by extension at 72 °C for 10 min. PCR product quality was assessed with 1% agarose gel electrophoresis. PCR products were cleaned using the QIAquick PCR purification kit (Qiagen, Valencia, CA), quantified using a Synergy HT microplate reader (Biotek, Winooski, Vermont, USA), and digested with MspI. After digestion, a cocktail was made containing 0.5 µl LIZ labeled GeneScan 1200 internal size standard (Applied Biosystems, Grand Island, NY), 12.5 µl Hi–Di formamide (Applied Biosystems, Grand Island, NY), and 1 µl of digested product which was centrifuged, then incubated at 94 °C for 4 min followed by incubation at 4 °C for 5 min. Fragments were analyzed on an ABI Prism 3100 genetic analyzer (Applied Biosystems, Grand Island, NY).

TRFLP profiles were measured using the GeneMapper software (Applied Biosystems, NY) with a cutoff of 50 bp. The relative abundance of a TRF in a TRFLP profile was calculated by dividing the peak height of the TRF by the total peak height of all TRFs in the profile (Singh et al., 2006). Community analyses of fragments were conducted using Primer 6 with site specified as a factor and soil temperature and soil moisture specified as covariates (Primer-E Ltd., United Kingdom). Soil temperature and soil moisture were significantly different between the southern and northern site (soil temperature, F = 187.8, p < 0.01; soil moisture, F = 17.6, p < 0.01). Thus, we followed up the community analyses and separated the data by site using a distance based linear model (DISTLM) that assessed the effect of soil temperature and soil moisture on total microbial, fungal, and bacterial community composition at each site (Anderson, 2004; Langlois, Anderson & Babcock, 2006). Additionally, bacterial, fungal, and total microbial richness for all chambers at each site was calculated by summing the unique number of TRFs in each sample.

We assayed microbial activity by measuring potential extracellular enzyme activity using methylumbelliferone (MUB) linked substrates and 3,4 Dihydroxyphenylalanine (L-DOPA) (Saiya-Cork, Sinsabaugh & Zak, 2002). Soils were assayed for nine ecologically relevant enzymes in order to assess the functional diversity of the soil community: sulfatase (hydrolysis of sulfate esters), nitrogen acetylglucosaminidase (nagase; mineralization of nitrogen from chitin), xylosidase (hemicellulose degradation), phosphatase (hydrolysis of phosphomonoesters and phosphodiesters releasing phosphate), α-glucosidase (degradation of storage carbohydrates), β-glucosidase (degradation of cellulose and other–1,4 glucans), cellobiohydrolase (cellulose degradation), phenol oxidase (lignin degradation), and peroxidase (lignin degradation). Soils were prepared by adding 125 ml of 0.5 M sodium acetate buffer (buffer, pH 5) to approximately 1 g of soil and homogenized for 2 min by immersion blending. Sulfatase, nagase, xylosidase, phosphatase, α-glucosidase, β-glucosidase, and cellobiohydrolase were measured using MUB linked substrates. We prepared 96 well plates with blanks, experimental controls, and samples, which were replicated 8 times each. All plates were incubated at room temperature in the dark. The nagase and phosphatase reactions were incubated for 0.5 h, while sulfatase, xylosidase, α-glucosidase, β-glucosidase, and cellobiohydrolase were incubated for 2 h. Fluorescence was read at an excitation of 365 nm and an emission of 450 nm (Biotek, Winooski, Vermont, US). Phenol oxidase and peroxidase activity were measured using L-DOPA. Assays were replicated 16 times and reactions were incubated in the dark for 24 h. Absorbance was read at 460 nm on a Synergy HT microplate reader (Biotek, Winooski, Vermont, US). Potential enzymatic activity is presented as nmol h−1 g−1 (Saiya-Cork, Sinsabaugh & Zak, 2002; Sinsabaugh, 1994).

The decomposition rate of a standard cellulose substrate was measured in each chamber to determine how warming might alter the rate of carbon degradation, a microbially mediated process. Twelve mesh decomposition bags (10 cm × 10 cm; 3 mm mesh double layered on top and 1.3 mm mesh on bottom) containing 10 g of Whatman # 1 filter paper were deployed in each of the chambers and collected after 3, 6, 9, and 12 months. All data are shown on an ash-free oven dry mass basis. K-constants were calculated for each chamber at each site following collection (Olson, 1963).

Because microbial communities experience changes in soil temperature and moisture as a result of changing air temperature, we used an analysis of covariance (ANCOVA) to examine the effect of site, average organic layer soil temperature on the day of sampling, average volumetric soil moisture on the day of sampling, and the interactions of these factors on microbial community composition, abundance, enzymatic activity, and decomposition rates. When three way interactions among site, soil temperature, and soil moisture were detected, we separated the data by site and ran regressions using soil temperature and soil moisture as factors. We assessed the effect of minimum, maximum, and variation in soil temperature and moisture over one year on microbial structure and function, but found no significant effects, so results including those factors are not presented.

Results and Discussion

Due to their short generation times and rapid turnover, soil bacteria are predicted to respond quickly to global warming (Pietikainen, Pettersson & Baath, 2005; Rinnan et al., 2007; Zogg et al., 1997). Soil fungi, in contrast, are relatively slow growing, so fungal responses may lag relative to bacteria and be a function of substrate availability (de Boer et al., 2005). Consistent with this, we found that experimental warming influenced the bacterial communities to a greater extent than the fungal communities at the southern site (Fig. 1, Appendix 4). Optimal soil temperatures for bacterial growth range between 25 and 30 °C (Pietikainen, Pettersson & Baath, 2005)—a temperature that is much higher than the soil temperatures measured on the days we collected our samples (14 and 9.5 °C at the southern and northern site, respectively). The observed temperatures in chamber soils are well below the optimum for bacterial growth, thus warming should increase both bacterial abundance and function at both sites. Surprisingly, bacterial abundance increased with warmer temperatures only at the southern site, not at the northern site. Bacterial abundance responded to a significant site × soil temperature × soil moisture interaction (F1,16 = 18.17, P < 0.01). Bacterial abundance was greatest when soil moisture was low and soil temperatures were high at the southern site (Fig. 1A, F1,8 = 16.11, P < 0.01). However, there was no effect of soil temperature or moisture on bacterial abundance at the northern site (Fig. 1B, F1,8 = 0.86, P = 0.50), where soils were cooler, moister, and fungal dominated.

Figure 1 Site, soil temperature, and soil moisture interactively altered bacterial abundance (F = 18.17, p < 0.01).

(A) At the southern site, abundance was greatest at low soil moisture and high soil temperatures. (B) At the northern site, there was no effect of soil temperature or moisture on abundance.

Figure 2 Fungal abundance was similar among sites, soil temperatures, and soil moistures at the southern (A, F = 0.56, p = 0.48) and northern site (B, F = 1.86, p = 0.21).

Fungal abundance was 3.5× greater at the northern site than at the southern site indicating a fungal dominated decomposition pathway (Fig. 2). Because soil fungi and bacteria compete for resources, at the northern site fungi may outcompete bacteria for resources suppressing bacterial growth, in essence preventing the bacteria from responding to warming. Across treatments, bacterial abundance was 1.6× higher at the southern site relative to the northern site (F1,16 = 9.22, P = 0.01), indicating a bacterial dominated decomposition pathway at the southern site (Allison & Treseder, 2008; Strickland & Rousk, 2010). In contrast, we found that bacterial richness was highest at intermediate soil temperatures and soil moistures at the southern site (F1,16 = 23.51, P < 0.01). Importantly, these differences between sites match predictions that the responses of organisms to changes in temperature vary between southern and northern sites. Alternatively, changes in the richness and abundance of predatory organisms like soil nematodes may constrain bacterial abundance in response to warming (Briones et al., 2009).

In addition to the effects of warming on microbial abundance and diversity there was a significant effect of soil temperature on total microbial community composition (F1,16 = 17.86, P < 0.01), fungal community composition (F1,16 = 4.33, P < 0.01), and bacterial community composition (F1,16 = 41.89, p < 0.01). Additionally, total microbial community composition (F1,16 = 1.67, P = 0.03) and bacterial community composition (F1,16 = 3.43, P < 0.01) differed between sites. When sites were analyzed separately, there was no effect of soil temperature or moisture on community composition. This suggests that the effect of soil temperatures on community composition was driven by large landscape-scale differences in soil temperature, and other unmeasured factors such as differences in plant community composition or historical legacies, between the two sites rather than any differences among treatments within sites.

We predicted soil enzymatic activity would proportionally increase with temperature because enzyme reaction rates are temperature sensitive. Instead, we found that site, soil temperature, and soil moisture interacted to influence enzyme activity. The highest levels of xylosidase activity—an enzyme involved in hemicellulose degradation—were at intermediate temperatures and low levels of soil moisture at the southern site (Fig. 3A, F1,8 = 29.57, P < 0.01), but xylosidase activity did not differ among treatments at the northern site. The low levels of soil moisture at the southern site may have decreased enzymatic turnover leading to an increase in “standing”, but not active enzyme (Wallenstein & Weintraub, 2008). When soil moisture levels were low, extracellular enzymes are more likely to be adsorbed onto soil particles making them relatively inactive in situ, even though they produced higher measurable activity in the laboratory (Wallenstein & Weintraub, 2008). At the northern site, nagase activity was highest at intermediate levels of soil moisture and high soil temperatures (Fig. 3D, F1,8 = 5.25, P = 0.05), but there was no effect of treatment on nagase activity at the southern site (Fig. 3C, F1,8 = 1.19, P = 0.31), suggesting that nitrogen limitation may be greater at the northern site during this time.

Figure 3 Site, soil temperature, and soil moisture interactively altered potential xylosidase (F = 10.22, p = 0.01) and nagase (F = 5.42, p = 0.03) activity.

(A) At the southern site, xylosidase activity was lowest under high soil temperatures and low soil moisture. (B) At the northern site, there was no effect of soil temperature or moisture on xylosidase activity. (C) At the southern site, there was no effect of soil temperature or moisture on nagase activity. (D) At the northern site, nagase activity was greatest under high soil temperatures and moisture.

Table 1 Soil temperature and moisture independently and interactively altered microbial community structure and function.

F and p statistics, in parentheses, show the main effects of soil temperature and soil moisture and the soil temperature × moisture interactions within each site. P values ≤ 0.05 are shown in bold. A distance based linear model (DISTLM) was used to assess community composition thus F and p values were not obtained for the full model or the interaction term.

	Southern site	Northern site	
	Full
model	Soil
temperature	Soil
moisture	Soil temp ×
soil moisture	Full
model	Soil
temperature	Soil
moisture	Soil temp ×
soil moisture	
Microbial community
composition	na	0.75 (0.81)	0.83 (0.64)	na	na	0.83 (0.72)	1.05 (0.39)	na	
Fungal community
composition	na	0.87 (0.66)	0.98 (0.48)	na	na	0.62 (0.89)	1.33 (0.18)	na	
Bacterial community
composition	na	0.74 (0.74)	0.89 (0.58)	na	na	0.97 (0.47)	0.62 (0.86)	na	
Total richness	2.67 (0.12)	4.77 (0.06)	0.55 (0.48)	3.12 (0.12)	0.68 (0.59)	0.44 (0.53)	0.62 (0.45)	1.59 (0.24)	
Fungal richness	3.25 (0.08)	4.20 (0.07)	1.35 (0.28)	5.04 (0.05)	0.21 (0.89)	0.04 (0.85)	0.11 (0.75)	0.56 (0.48)	
Bacterial richness	11.18 (<0.01)	1.30 (0.29)	17.29 (<0.01)	23.51 (<0.01)	1.04 (0.43)	1.01 (0.34)	1.15 (0.32)	2.03 (0.19)	
Fungal:bacterial	1.64 (0.26)	0.03 (0.87)	0.38 (0.55)	4.86 (0.06)	0.58 (0.66)	0.02 (0.90)	1.12 (0.32)	0.56 (0.48)	
Fungal abundance	0.38 (0.77)	0.02 (0.90)	0.86 (0.38)	0.56 (0.48)	0.79 (0.52)	0.37 (0.56)	0.72 (0.42)	1.86 (0.21)	
Bacterial abundance	5.40 (0.03)	0.002 (0.97)	0.64 (0.45)	16.11 (<0.01)	0.86 (0.50)	1.86 (0.21)	0.17 (0.69)	0.97 (0.35)	
Xylosidase	10.39 (<0.01)	1.98 (0.20)	0.61 (0.46)	29.57 (<0.01)	0.68 (0.59)	1.79 (0.22)	0.13 (0.73)	0.13 (0.72)	
Sulfatase	0.52 (0.68)	0.32 (0.59)	0.08 (0.79)	1.28 (0.29)	2.57 (0.13)	0.003 (0.96)	7.42 (0.03)	0.03 (0.86)	
Cellobiohydrolase	1.49 (0.29)	1.57 (0.25)	1.78 (0.22)	1.47 (0.26)	0.16 (0.92)	0.40 (0.55)	0.03 (0.87)	0.0000 (0.99)	
β-glucosidase	1.56 (0.27)	0.70 (0.43)	1.62 (0.24)	3.12 (0.12)	0.88 (0.49)	1.89 (0.21)	0.34 (0.58)	0.63 (0.45)	
α-glucosidase	0.72 (0.57)	0.50 (0.50)	1.43 (0.27)	0.05 (0.83)	1.37 (0.32)	0.32 (0.59)	0.82 (0.39)	2.87 (0.13)	
Nagase	0.50 (0.70)	0.01 (0.94)	0.63 (0.45)	1.19 (0.31)	2.06 (0.18)	0.62 (0.45)	0.48 (0.51)	5.25 (0.05)	
Phosphatase	0.97 (0.45)	0.22 (0.65)	0.52 (0.49)	1.38 (0.27)	1.46 (0.30)	3.13 (0.12)	0.11 (0.75)	2.64 (0.14)	
Phenol oxidase	3.50 (0.07)	7.63 (0.02)	1.76 (0.22)	2.14 (0.18)	1.53 (0.28)	1.98 (0.20)	1.87 (0.21)	0.69 (0.43)	
Peroxidase	na	na	na	na	1.85 (0.22)	4.91 (0.06)	0.0004 (0.98)	0.001 (0.97)	
Decomposition
(k constant)	0.01 (0.10)	0.02 (0.90)	0.004 (0.95)	0.01 (0.92)	2.10 (0.18)	4.67 (0.06)	1.01 (0.34)	0.12 (0.73)	

Interestingly, the changes in microbial community composition, abundance, and potential activity did not influence cellulose decomposition between the sites or among the warming treatments (Table 1, P > 0.05, Appendix 1). One possibility is that decomposer communities are functionally redundant, thus shifts in their communities due to warming do not alter decomposition (Allison & Martiny, 2008; Allison & Treseder, 2008). Because this experiment had been running for only a year, it is more likely that the communities are still using up carbon substrates present in the soil prior to plot establishment and that changes in the decomposition process may take longer than one year before they become evident (Rinnan et al., 2007). Nevertheless, the changes in the bacterial community and potential microbial function in response to experimental warming suggest that as soil temperatures increase, changes in the fungal community and, at least some of the processes it mediates will follow. But, just as importantly, these effects need not be consistent among sites, which suggests that models predicting ecosystem responses to climate change should account for geographic variation in responses to ongoing warming.

Conclusions

We conclude that the effect of warming on microbial community structure and function may become more pronounced as soil temperatures increase and carbon substrates are depleted through time. Further the response of communities, even communities in the same ecosystem, will likely vary by location. Thus, predicting and modeling the extent to which terrestrial ecosystems will respond to global change requires globally replicated experiments across habitat types and climates.

Supplemental Information

Supplemental Information 1 Appendices 1–4

Click here for additional data file.

We thank S Pelini, M Pelini, L Nichols, C Patterson, and L Breza for help with sample collection and processing. We thank N Gotelli and A Ellison for establishing and maintaining the warming chambers at Harvard Forest.

Additional Information and Declarations

Competing Interests

Author Contributions

Nathan J. Sanders is an Academic Editor for PeerJ.

Melissa A. Cregger conceived and designed the experiments, performed the experiments, analyzed the data, contributed reagents/materials/analysis tools, wrote the paper, prepared figures and/or tables, reviewed drafts of the paper.

Nathan J. Sanders conceived and designed the experiments, performed the experiments, analyzed the data, wrote the paper, reviewed drafts of the paper.

Robert R. Dunn conceived and designed the experiments, wrote the paper, reviewed drafts of the paper.

Aimée T. Classen conceived and designed the experiments, performed the experiments, analyzed the data, contributed reagents/materials/analysis tools, wrote the paper, reviewed drafts of the paper.

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
