# Peer review of "Microbial communities respond to experimental warming, but site matters"

_PeerJ, doi:10.7717/peerj.358_

## Round 0.1 · original submission · Major Revisions

Your appeal convinced me to review this paper more carefully and I sought out a very respected microbiologst for another review. He also had major problems with the paper so you have a lot of work to do to make it publishable in PeerJ.

Reviewer 1 ·

Basic reporting

Earlier studies of microbial and animal responses to warming are cited, and the authors hypothesized that there would be a more noticeable response to warming at the southern site (Duke forest) than at the northern (Harvard forest). I would make Appendix 4 a figure, since results for fungi are no less important than those for bacteria.
l. 64. “Materials and Methods” to agree with journal style.
There is no Conclusions section, although the Journal style suggests there should be one.
l. 270-274. The Acknowledgements section should not contain funding information,
l. 274. “References”, not “Literature Cited”, as per journal style.

Experimental design

The study was designed to assess whether bacterial community structure and function would respond differently to soil warming at a site near the northern and southern boundaries of the eastern deciduous forest. At each site open top chambers (OTCs) allow for increases to ambient air temperature at 0.5°C steps between 1.5 and 5.5°C above ambient. Please indicate chamber dimensions and that there was a tree in the center of each. Samples were taken from each chamber on one date in spring of 2011. Dates should be given for these collections. I assume this means (5 cores/chamber x 9 steps/site x 2 sites = 90 samples, but this should be explicitly stated). In general the Methods section is clear. However, I am not certain how the soil moisture data are used in the analyses. This parameter was monitored continuously (l. 95), but what values are employed in the analyses to generate the gradient shown in the figures. Is this an annual value, a monthly value, the value during the week before the single sampling date? Please explain and justify the appropriateness of the values chosen.

Validity of the findings

The Results and Discussion section is clearly written, but I have some questions regarding interpretation of results and discussion. For example, the descriptors “quickly” and “slow growing” are used to characterize bacterial and fungal growth. I get the point, but the chambers were initiated in January 2010 and you sampled sometime in the spring of 2011. It would seem that the differences in growth rates would be most apparent on a time scale of days – weeks, not 1 + year. After allowing for some expected seasonality, that the condition at the time of sampling presumably represented some sort of equilibrium condition. My biggest concern is the statement that “warming should increase both bacterial abundance and function at both sites” (l. 210-211). One would expect the effect on function, but abundance is impacted by both bottom-up (nutrient supply, temperature, etc.) factors and top-down (grazing) factors. Attention should be given to densities of protozoa, meiofauna, and macrofauna in the soils and how their grazing of the bacterial and fungal communities could factor into the interpretation of results or the authors should substantiate why grazing does not impact their findings.

Additional comments

Some other detailed comments:
l. 18. “scale-up” Overused jargon. Rephrase.
l. 26. Insert “community” to read “microbial community structure”
l. 40. I don’t think you mean “northern and southern range” but rather “extremes of the range”
l. 44. I suggest “Work at these sites and elsewhere indicates …”
l. 62. I think that including the explanatory phrase about fungi here only muddies the point.
l. 124 – 125. Italicize taxa names.
l. 140. Degree sign needed after the first 95.
l. 165. Why is sulfate underlined?
l. 196 – 199. This is a result and should be moved to somewhere in that section.
l. 218. “3.5 x greater” than what? than where? Do you mean than at the southern site, or than bacterial abundance? Clarify.

Reviewer 2 ·

Basic reporting

It is unclear how many samples are analyzed. The text says 12 open-top-chambers (line 65), but then says temps from 1.5 to 5.5C over ambient were analyzed at 0.5C increments (9 samples). Are there 12 otc’s at each site or 12 otc’s total; this is unclear. Either way, the numbers (9 temp treatments / 12 otc’s) don’t add up. The number of samples is never reported. (Acutally, I now see that this may be reported in Appendices 2 and 3 – As printed, I only see some of the columns of these tables.) Either way, the number of samples in unclear.

Experimental design

I don't understand why five soil cores were taken from each open-top-chamber and combined. It makes perfect sense to combine cores from a site when a large number of sites are being analyzed, but this study uses a very small number of sites. By analyzing each core separately (for bacterial/fungal abundance, community composition, and enzyme activity), some estimate of within-site variation could have been made. The cost-effectiveness is not a valid argument -- the methods used in this manuscript are low cost.

There is absolutely no reason to be using TRFLPs to describe and compare microbial communities in 2013. The cost of DNA sequencing is absurdly cheap.

Figures 1 and 2 are pretty, but appear largely driven by your very small sample size. This is especially true for Figure 1A, in which one soil (warm, dry) has greater abundance of 16S than other samples.

In short, I have major problems with your small sample size (10 samples per site? – again, it is unclear) and your characterizations of the microbial community using outdated technology.

Validity of the findings

The findings are not valid for two main reasons: 1) The sample size is very small (10 samples per site / no replication within temperature treatments), and 2) Microbial communities are characterized using TRFLPs.

Additional comments

If you think you want to pursue this study further, I would strongly recommend characterizing the microbial communities using DNA sequence data. It's a shame that you combined the 5 cores sampled within each OTC -- if samples from each core exist separately, it would be very useful to use those to estimate the variation in richness, diversity, abundance, etc. within OTC sites.

---

## Round 0.2 · accepted · Accept

I think you addressed the comments of the reviewers sufficiently well. Thanks for your attention to the details of the revision.